# Three-dimensional reconstruction of a whole insect reveals its phloem sap-sucking mechanism at nano-resolution

**Xin-Qiu Wang[1†], Jian-sheng Guo[2†], Dan-Ting Li[1,3], Yang Yu[4], Jaco Hagoort[5], Bernard Moussian[6], Chuan-Xi Zhang[3]\***

[1]Institute of Insect Science, Zhejiang University, Hangzhou, China; [2]Department of Pathology of Sir Run Run Shaw Hospital, and Center of Cryo-Electron Microscopy, School of Medicine, Zhejiang University, Hangzhou, China; [3]State Key Laboratory for Managing Biotic and Chemical Threats to the Quality and Safety of Agro-Products, Key Laboratory of Biotechnology in Plant Protection of Ministry of Agriculture and Zhejiang Province, Institute of Plant Virology, Ningbo University, Ningbo, China; [4]Carl Zeiss (Shanghai) Co., Ltd.60 Meiyue Road, China (Shanghai) Pilot Free Trade Zone, Shanghai, China; [5]Department of Medical Biology, Amsterdam University Medical Centers, University of Amsterdam, Amsterdam, Netherlands; [6]Université Côte d'Azur, CNRS, Université Côte d'Azur, Institute of Biology Valrose, Parc Valrose, Inserm, France

**Abstract** Using serial block-face scanning electron microscopy, we report on the internal 3D structures of the brown planthopper, *Nilaparvata lugens* (Hemiptera: Delphacidae) at nanometer resolution for the first time. Within the reconstructed organs and tissues, we found many novel and fascinating internal structures in the planthopper such as naturally occurring three four-way rings connecting adjacent spiracles to facilitate efficient gas exchange, and fungal endosymbionts in a single huge insect cell occupying 22% of the abdomen volume to enable the insect to live on plant sap. To understand the muscle and stylet movement during phloem sap-sucking, the cephalic skeleton and muscles were reconstructed in feeding nymphs. The results revealed an unexpected contraction of the protractors of the stylets and suggested a novel feeding model for the phloem sap-sucking.

**\*For correspondence:** chxzhang@zju.edu.cn

[†]These authors contributed equally to this work

## Introduction

The morphology and arrangement of organs in the insect body have been studied as early as in the nineteenth century (e.g *Lowne, 1892*). These data are still used in current textbooks and atlases of insect morphology and physiology. However, the internal structures and their spatial relationship lack a direct three-dimensional (3D) representation. In recent years, morphological studies using micro-computed tomography (micro-CT) served to reveal the internal structures of some insect tissues, such as the copulating *Drosophila* (*Mattei et al., 2015*) or even of the entire organism such as the *Drosophila* pupa (*Schoborg et al., 2019*). However, for small insects and delicate structures, the resolution of micro-CT is insufficient to distinguish details. Serial block-face scanning electron microscopy (SBF-SEM) allows visualization of the insect's internal structure at the nanometer resolution in 3D. This method relies on reiterated imaging and sectioning of resin embedded sample using a robotic ultramicrotome and digital assembly of the single images to 3D images (*Denk and Horstmann, 2004*). Several 3D structures of insect tissues have been resolved by SBF-SEM, such as pericardial nephrocytes in *Drosophila* (*Kawasaki et al., 2019*), and cyst-like bodies formed by

**eLife digest** Since the 19th century, scientists have been investigating how the organs of insects are shaped and arranged. However, classic microscopy methods have struggled to image these small, delicate structures. Understanding how the organs of insects are configured could help to identify new methods for controlling pests, such as chemicals that target the mouthparts that some insects use to feed on plants.

Most insects that feed on the sap of plants suck out the nutrient via their stylet bundle – a thin, straw-like structure surrounded by a sheath called the labium. As well as drying out the plant and damaging its tissues, the stylet bundle also allows the insect to transmit viruses that cause further harm. To investigate these mouthparts in more detail, Wang, Guo et al. used a method called SBF-SEM to determine the three-dimensional structure of one of the most destructive pests of rice crops, the brown planthopper.

In this technique, a picture of the planthopper was taken every time a thin slice of its body was removed. This continuous slicing and re-imaging generated thousands of images that were compiled into a three-dimensional model of the brown planthopper's whole body and internal organs. Previously unknown features emerged from the reconstruction, including a huge cell in the planthopper's abdomen which is full of fungi that provide the nutrients absent in plants.

Next, Wang, Guo et al. used this technique to see how the muscles in the labium and surrounding the stylet move by imaging planthoppers that were frozen at different stages of the feeding process. This revealed that when brown planthoppers bow their heads to eat, the labium compresses and pushes out the stylet, allowing it to pierce deeper into the plant.

This is the first time that the body of such a small insect has been reconstructed three-dimensionally using SBF-SEM. Furthermore, these findings help explain how brown planthoppers and other sap-feeding insects insert their stylet and damage plants, potentially providing a stepping stone towards identifying new strategies to stop these pests from destroying millions of crops.

*Trypanosoma brucei* in the anterior midgut of tsetse flies (*Rose et al., 2020*). Insects of small size are suitable for this technology to obtain an integrated 3D atlas of the whole body with a resolution higher than possible by any other previous method. Resolving the internal structures and their spatial relationship is of great value in understanding the relationship of structure and function, and in conducting physiological, biochemical, and molecular studies in this direction. Although reconstruction of a 3D atlas of the whole body of any insect by EM method is valuable and theoretically possible, it has not been conducted yet.

Hemipteran insects are in general small and possess delicate mouthparts with straw-like stylets that are highly specialized for penetrating plant tissues. For virus vectors from the order of Hemiptera, including small/tiny insects like planthoppers, leafhoppers, psyllids, aphids, and whiteflies, studying their phloem sap-sucking mechanism will enable us to understand the way they injure crops and inject their saliva with transmitted viruses into crop tissues. Knowledge about their internal structures such as the mouthparts and alimentary canal of the phloem sap-sucking *Diaphorina citri* (Hemiptera: Liviidae) (*Ammar et al., 2017*), *Psammotettix striatus* (Hemiptera: Cicadellidae) (*Zhang et al., 2012*), and the aphid *Acyrthosiphon pisum* (Hemiptera: Aphididae) (*Guschinskaya et al., 2020*) is restricted to studies that relied on some discrete images by either light, confocal, or electron microscopy. Study of phloem–sucking insect/pathogen interactions represents an exciting frontier of plant science (*Jiang et al., 2019*). When sucking in plant sap, hemipterans are proposed to protrude their straw-like stylets into the plant tissue by contracting the respective muscles (*Leopold et al., 2003*; *Beardsley and Gonzalez, 1975*), but an evidence of the muscle movement has not been provided. Indeed, the commonly assumed feeding mechanism for phloem sap-sucking insects was largely speculative instead of being based on the actual movement of muscles and the feeding apparatuses.

The rice sap-sucking brown planthopper (BPH), *Nilaparvata lugens* (Stål) (Hemiptera: Fulgoroidea: Delphacidae), is the most important pest of the rice plant. They damage rice not only by directly feeding and ovipositing on it but also by transmitting two rice viruses, rice rugged stunt virus and rice grassy stunt virus. Dense BPH infestations can cause complete drying and wilting of rice plants,

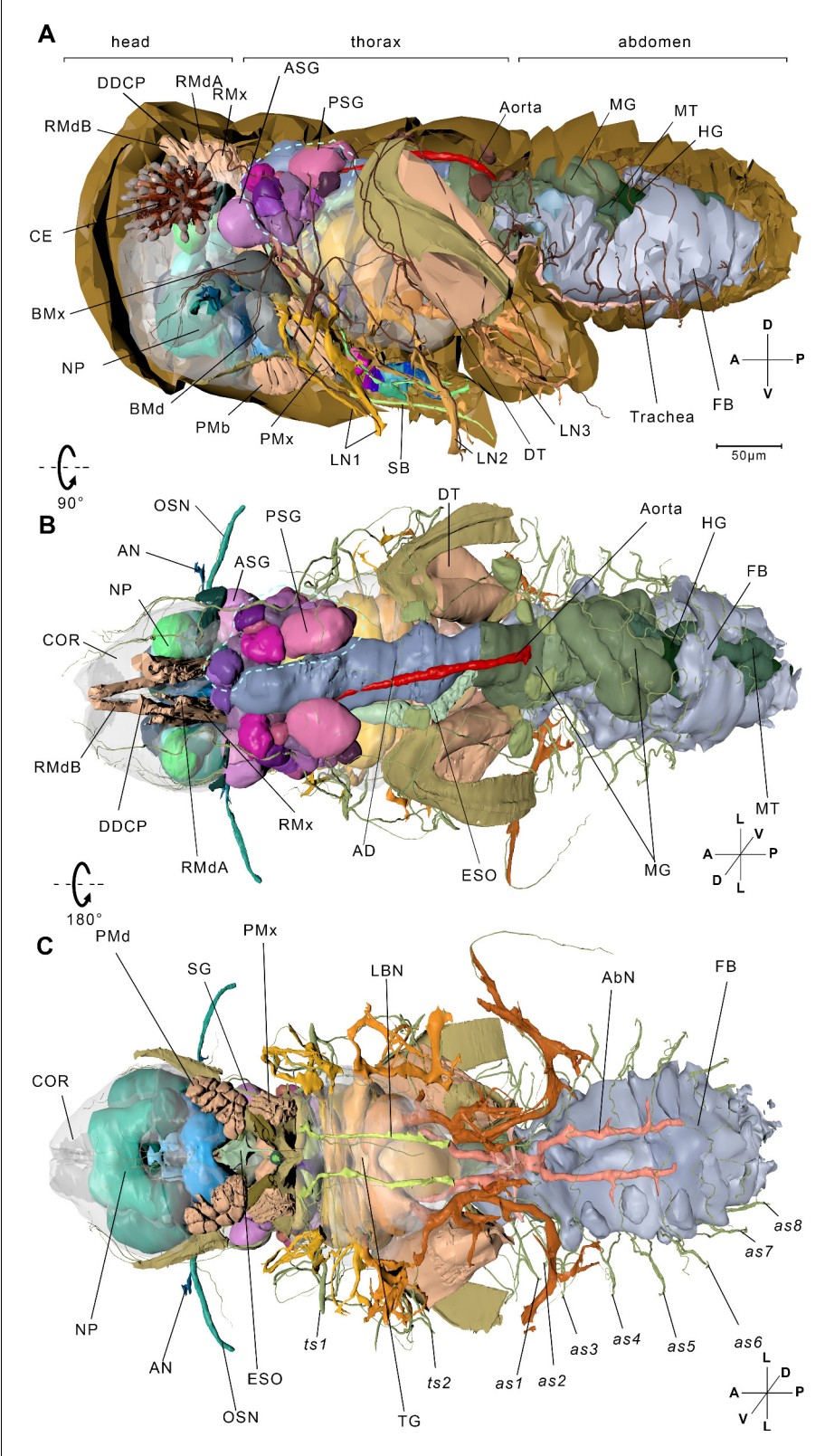

**Figure 1.** Reconstructed internal structures of a first instar nymph. (**A**) Left-lateral view. Half of the exoskeleton is shown. The body can be grouped into three parts, the head, thorax, and abdomen. The green dotted line indicates principle salivary gland (PSG). (**B**) Dorsal view. All exoskeleton is removed. (**C**) Ventral view. AbN, abdominal nerve; AD, anterior duct; AN, antennal nerve; as1–as8, spiracles on the first to eighth abdominal segments; ASG, accessory salivary gland; BMd, base of the mandibular stylet; BMx, base of the maxillary stylet; CE, compound eye; COR, cortex; DDCP, dorsal dilator

*Figure 1 continued on next page*

*Figure 1 continued*

of the cibarial pump; DT, depressor of the trochanter; ESO, esophagus; FB, fat body; HG, hindgut; LBN, labial nerve; LN1, proleg nerve; LN2, mesoleg nerve; LN3, metaleg nerve; MG, midgut; MT, Malpighian tubule; NP, neuropil; OSN, olfactory sensory nerve; PMd, protractor of the mandibular stylet; PMx, protractor of the maxillary stylet; RMdA and RMdB, retractor of the mandibular stylet A and B; RMx, retractor of the maxillary stylet; SB, stylet bundle; TG, thoracic ganglion; ts1, spiracle on the mesothoracic segment; ts2, spiracle on the metathoracic segment. See reconstructed central nervous system in *Figure 1—figure supplement 1*, serial block-face scanning electron microscopy (SBF-SEM) data of the whole insect in *Figure 1—video 1*, SBF-SEM data of the nervous system in *Figure 1—video 2* and reconstructed systems in *Figure 1—video 3*. Axis: D, dorsal; V, ventral; A, anterior; P, posterior; M, medial; L, lateral.

The online version of this article includes the following video and figure supplement(s) for figure 1:

**Figure supplement 1.** Reconstructed central nervous system and the compound eye.

**Figure 1—video 1.** The volume data of sample 1.

https://elifesciences.org/articles/62875#fig1video1

**Figure 1—video 2.** The volume data of sample 2.

https://elifesciences.org/articles/62875#fig1video2

**Figure 1—video 3.** The three-dimensional presentation of different systems and musculature based on the reconstructed model of sample 1.

https://elifesciences.org/articles/62875#fig1video3

a condition known as 'hopperburn'. Since the late 1960s, BPH outbreaks have been re-occurring approximately every 3 years in Asian countries, with their annual outbreak area amounting to approximately 10–20 million hectares of rice, resulting in millions of tons of losses. BPHs seasonally migrate from their overwintering rice fields in tropical Asia northward to temperate areas in China, Korea, Japan, and northern India when rice becomes available, inflicting damage in an even larger area across Asia (*Perfect and Cook, 1994*). BPH has been a model insect for ecological research and a model insect for hemipteran insects in biological research in recent years. In this study, we applied SBF-SEM to investigate the structure of various systems involved in food ingestion, digestion, locomotion, gas exchange and perception, and muscle movement during the feeding process of the nymph. A detailed 3D structure for a whole insect is presented for the first time, and a plant phloem sap-sucking model based on the actual movement of muscles and all feeding apparatuses during feeding process is proposed.

## Results

The readers are encouraged to read the results along with the interactive 3D images in the 3D PDF file, in which the readers can rotate the 3D, zoom in/out, hide or show any interested internal part of the insect in detail.

### 3D structures of the whole insect body

The length of the insect measures approximately 600 μm (*Figure 1*). We reconstructed the 3D structure of the whole insect body (*Figure 1*, *Figure 1—video 1*, *Figure 1—video 3*, interactive 3D PDF file), including the organ systems of food ingestion and digestion, locomotion, gas exchange, and perception. We discover several novel and fascinating internal structures in BPH, such as 24 neuropils in the central nerve system (*Figure 1—figure supplement 1*), three four-way rings connecting the spiracles in adjacent segments to facilitate efficient gas exchange (*Figure 2D,H–J*), and fungal endosymbionts in a single huge mycetocyte occupying 22% of the abdominal volume to enable the host to thrive on a low-nutrient diet provided solely by rice (*Figure 2G*, *Figure 2—figure supplement 2D*, *Table 2*).

The alimentary canal starts with specialized mouthparts and includes the esophagus, the anterior diverticulum (AD) (*Figure 2—figure supplement 1D*), the midgut (*Figure 2—figure supplement 1E*), and the hindgut (*Figure 2—figure supplement 1F*) and ends with the anus (*Figure 2—figure supplement 1G*). The anterior diverticulum (AD) extends anteriorly into the head (*Figure 2A,B*, Interactive 3D-PDF). The midgut is divided into an anterior sac and a midgut loop (*Figure 2—figure supplement 1A,E*). The midgut loop is a narrow tube coiling into a cluster of loops. The anterior part of the loop coils into the inner loop, and the outer loop has its posterior end sticking into the center of the cluster. After that, the midgut descends posteriorly to fuse with the Malpighian tubules (*Figure 2B*). The inner surface of the whole midgut is covered by a dense layer of microvilli,

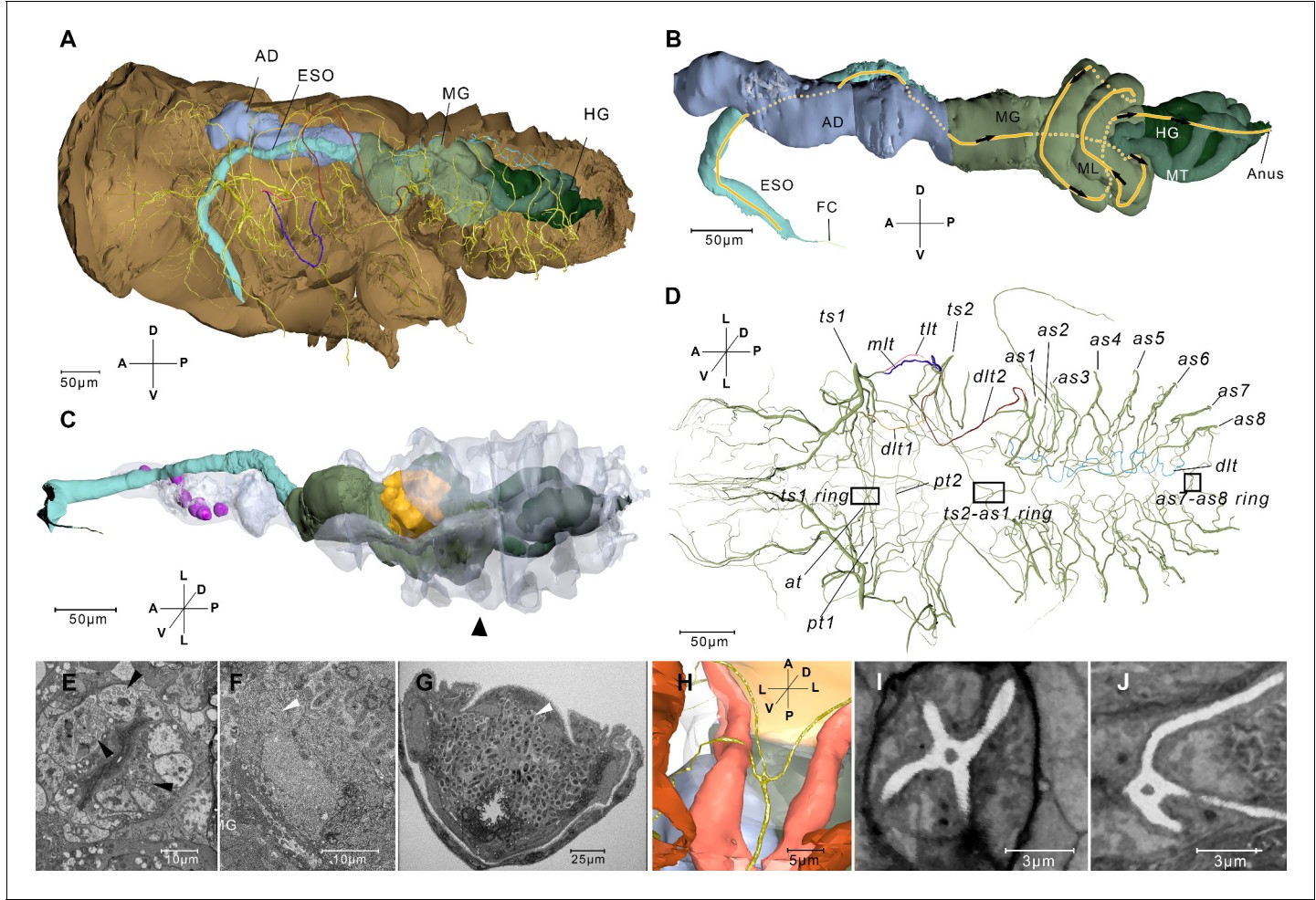

**Figure 2.** Reconstructed alimentary canal and tracheal system and endosymbionts. (**A**) The relationship of the tracheal system and the alimentary canal in the body. (**B**) Right-lateral view of the alimentary canal. The yellow line and the arrows indicate the direction in which the food can move in the midgut loop (ML). Details of the alimentary system are given in *Figure 2—figure supplement 2*. (**C**) The reconstructed alimentary canal and symbionts. Cells that accommodate symbionts in the anterior diverticulum (magenta), cells that accommodate thread-like symbionts (yellow), and the large mycetocyte that accommodate yeas-like symbionts (arrowhead) are shown. The anterior diverticulum and the large mycetocyte are rendered transparent. (**D**) Ventral view of the tracheal system. The rectangles indicate three four-way tracheal rings. (**E**) Anterior diverticulum symbionts are in different cells on the wall. The arrow heads indicate symbionts in three cells. Another image of this kind of symbiont is given in *Figure 2—figure supplement 2A*. (**F**) The thread-like symbionts are in a cell near the midgut (MG). The arrow head indicates the thread-like symbionts. The image of the thread-like symbionts in the midgut from another sample is given in *Figure 2—figure supplement 2C*. (**G**) A cross section of the abdomen and the yeast-like symbionts (arrowhead). A higher magnification image is given in *Figure 2—figure supplement 2D*. See serial block-face scanning electron microscopy (SBF-SEM) data of G in *Figure 2—video 1*. (**H**) Reconstructed ts2-as1 four-way tracheal ring (green). The detailed features of the spiracle are given in *Figure 2—figure supplement 3*. (**I and J**) Two slices reconstructed from the serial block-face scanning electron microscopy (SBF-SEM) data set showing the ts1–ts2 and the ts2–as1 four-way tracheal ring respectively. AD, anterior diverticulum; *as1–as8*, spiracles on the first to eighth abdominal segments; *at*, the trachea extending anteriorly from the tracheal vestibule of *ts1*; *dlt*, dorsal longitudinal trunk; *dlt1*, dorsal longitudinal trunk from *ts1*; *dlt2*, dorsal longitudinal trunk from *ts2*; ESO, esophagus; FC, food canal; HG, hindgut; MG, midgut; *pt1*, the trachea extending posteriorly from *ts1*; *mlt*, mesoleg trunk; *pt2*, the second trachea extending posteriorly from *ts1*; *tlt*, thorax lateral trunk; *ts1*, the spiracle on the mesothoracic segment; *ts2*, the spiracle on the metathoracic segment. Axis labels are the same as those used in *Figure 1*.

The online version of this article includes the following video and figure supplement(s) for figure 2:

**Figure supplement 1.** Reconstructed alimentary canal and serial block-face scanning electron microscopy (SBF-SEM) images.

**Figure supplement 2.** Serial block-face scanning electron microscopy (SBF-SEM) sections of symbionts and their positions in the nymph.

**Figure supplement 3.** Reconstructed spiracles and corresponding serial block-face scanning electron microscopy (SBF-SEM) images.

**Figure supplement 4.** The dissected alimentary canals of a female adult *N. lugens*.

**Figure supplement 5.** Phylogeny of higher taxa in the Hemiptera (*Johnson et al., 2018*).

**Figure 2—video 1.** The volume data of sample 5.

https://elifesciences.org/articles/62875#fig2video1

**Table 1.** Samples used for reconstruction.

| Sample | Body part | Slices | Pixel size (nm) | z-resolution (nm) |
|---|---|---|---|---|
| 1 | Whole body | 10,042 | 53 | 55 |
| 2 | A head with the prothorax | 4049 | 55 | 70 |
| 3 | A head with intact mouthpart | 2832 | 27 | 100 |
| 4 | Several abdominal segments | 2067 | 55 | 70 |
| 5 | Posterior end | 1700 | 55 | 35 |
| 6 | A head with the mouthpart (feeding) | 2720 | 35 | 100 |
| 7 | A head with the mouthpart (feeding) | 2891 | 37 | 100 |

especially in the loop region where the gut lumen is almost constricted (*Figure 2—figure supplement 1E*). There is no membrane or sheath enveloping the midgut loop, and the filter chamber, which is a common structure in sap-sucking Hemipteran insects, is not found (*Figure 2—figure supplement 4*). The anterior midgut contacts the end of the loop region at one point where the epithelia of the anterior midgut become thinner than in the neighboring area (*Figure 2—figure supplement 1E*). The midgut loop at position '3' is connected to the hindgut.

The 3D reconstruction reveals the distribution of three structurally different endosymbionts in the nymph. The most generally distributed endosymbiont is the Ascomycete fungus *Entomomyces delphacidicola* (*Fan et al., 2015*), also known as the yeast-like symbionts (YLSs), which have an ellipsoidal shape from a few microns to over 10 microns (*Figure 2G*, *Figure 2—figure supplement 2D*). Most of the YLS reside in a huge fat body mycetocyte that surrounds the midgut, taking up 22% of the abdominal volume and 4.2% of the total body volume (*Table 2*). A few YLSs are dispersed in the body cavity including the head and legs. Another two fat body mycetocytes adjacent to the midgut (*Figure 2C,F*) are full of thread-like bacterial endosymbionts. The anterior diverticulum lumen connects the midgut lumen though a narrow region (*Figure 2—figure supplement 1B*). Anterior to this narrow region, in the wall we found 10 hollow cells that each accommodates numerous rod-like bacterial endosymbionts (*Figure 2C,E*, *Figure 2—figure supplement 2A,B*), whereas in the anterior diverticulum lumen there are no endosymbionts. These cells seem to be the specialized structures to host the symbiont bacteria. 16S rDNA sequence of the AD showed that 95% of the symbionts are Arsenophonus species (*Figure 2—figure supplement 2E*).

The volume of total tracheae in the first instar nymph was 24,000 $\mu m^3$, accounting only for about 0.1% of the body volume (*Table 2*). Two pairs of thoracic spiracles (ts1 and ts2), eight pairs of abdominal spiracles (as1–as8), and tracheae extending from the spiracles make up the network (*Figure 2D*). Every abdominal spiracle is embedded in a pitting called the atrium with two valves, one of which separated the trachea and the atrium, while the other one divides the atrium into two parts (*Figure 2—figure supplement 3*). All the spiracles are connected by dorsal longitudinal trunks (dlt) (*Figure 2D*). At the ventral side, there are no longitudinal trunks. Three four-way tracheal rings at the ventral side connect the adjacent two spiracles and their counterparts at the other side of the body, like a traffic circle (*Figure 2D,H–J*). The ts2–as1 four-way tracheal ring and the as7–as8 four-way tracheal ring connect the spiracles in adjacent segments, as well as the left and right spiracles in the same segment. The as1 four-way tracheal ring connects two tracheae branches from the ts1 spiracle and their symmetrical counterparts at the opposite side.

In addition, we applied the SBF-SEM technique to explore the mechanism of phloem sap-sucking of *N. lugens*. Here, we focus on the sap-sucking process and the organs involved in food uptake.

## Mouthparts

The external morphology of the mouthparts in different Hemiptera is roughly identical. The beak is formed of the labium under a conical labrum (*Figure 3E*, *Figure 3—figure supplement 1C*, *Figure 3—figure supplement 2A,B*). The labium is 130 μm in length with three segments, but only the second (L2) and third segments (L3) are seen from the ventral side, as the first segment (L1) is covered by the labrum (*Figure 4A*, *Figure 3—figure supplement 2A,B*). A longitudinal groove on the ventral surface of the labium accommodates the stylet bundle, resulting in a longitudinal lumen. The

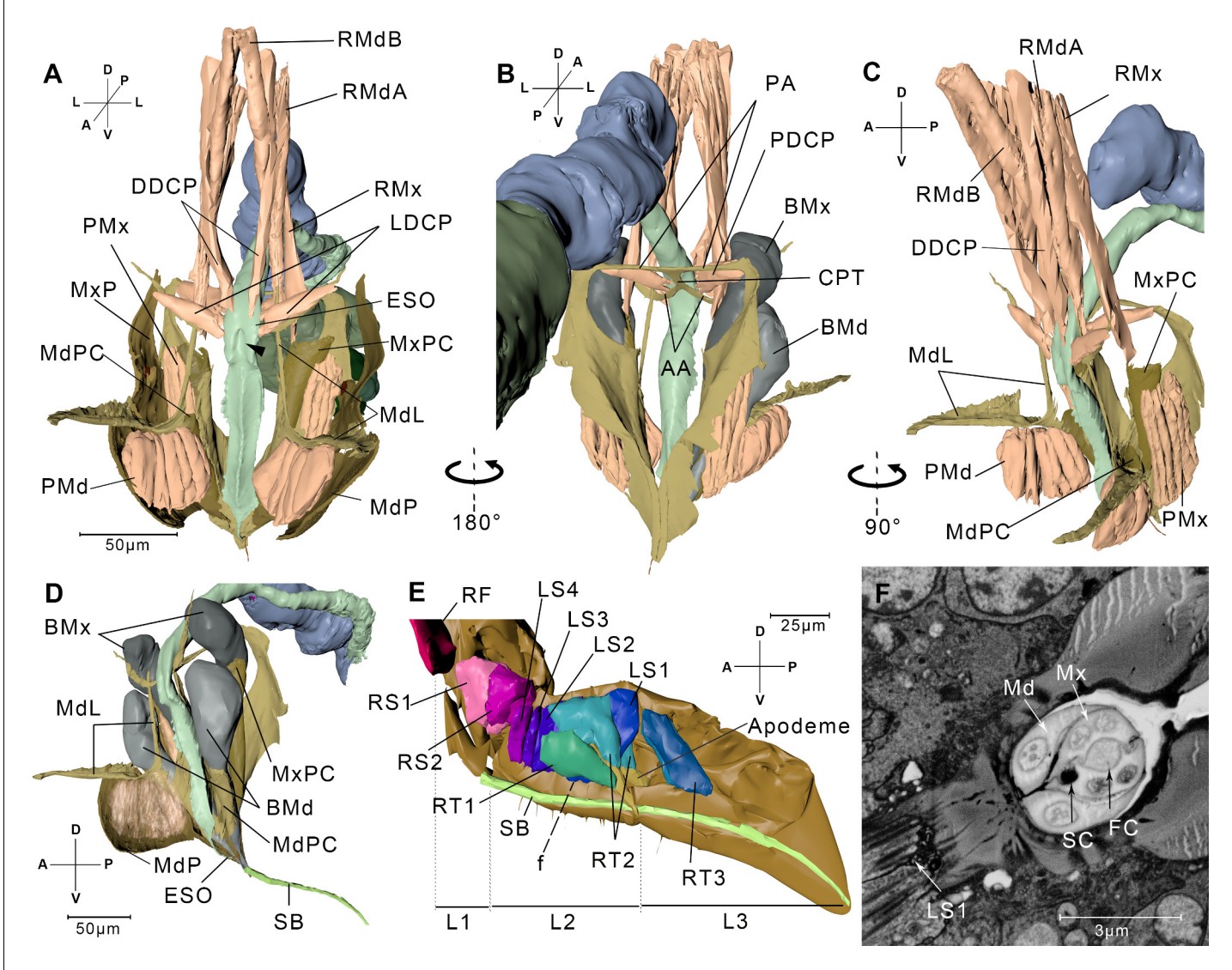

**Figure 3.** Reconstructed cephalic endoskeleton and mouthpart. (A) Front view of the cephalic endoskeleton and musculature. The arrowhead indicates the invagination on the anterior wall of the cibarial pump. The cephalic structures without the alimentary canal are in *Figure 3—figure supplement 1A, B*. (B) Back view of the cephalic endoskeleton and musculature. (C) Left-lateral view of the cephalic endoskeleton and musculature. (D) Left-lateral view of the junction between the base of the stylets and the stylet bundle. The muscles are rendered translucent, and the left mandibular lever with muscles is removed for a better view of the mandibular pouch (MdPC). (E) Left-lateral view of the mouthpart. Left half of the exoskeleton is removed. The overall view of the head and the mouthpart is in *Figure 3—figure supplement 1C*. (F) A slice from the serial block-face scanning electron microscopy (SBF-SEM) data set at the *f* position in E. AA, anterior arm; BMd, base of the mandibular stylet; BMx, base of the maxillary stylet; CPT, corpotentorium; DDCP, dorsal dilator of the cibarial pump; ESO, esophagus; FC, food canal; L1, L2, L3, The first, second, and third segments of the labium; LDCP, lateral dilator of the cibarial pump; LS1–4, lockers of the stylets 1–4; Md, mandibular stylet; MdL, mandibular lever; MdP, mandibular plate; MdPC, mandibular pouch; Mx, maxillary stylet; MxP, maxillary plate; MxPC, maxillary pouch; PA, posterior arm; PDCP, posterior dilator of the cibarial pump; PMd, protractor of the mandibular stylet; PMx, protractor of the maxillary stylet; RF, rotator of the first labial segment; RMdA, retractor of the mandibular stylet A; RMdB, retractor of the mandibular stylet B; RMx, retractor of the maxillary stylet; RS1–2, rotators of the second labial segment; RT1–3, retractors of the third labial segment 1–3; SB, stylet bundle; SC, saliva canal. See SBF-SEM data of the labium in *Figure 3—video 1*. Axis labels are the same as those used in *Figure 1*.

The online version of this article includes the following video and figure supplement(s) for figure 3:

**Figure supplement 1.** Cephalic endoskeleton and muscles and structures that are involved in the feeding process.

**Figure supplement 2.** Microscopic images of the mouthpart.

**Figure 3—video 1.** The volume data of sample 3.

https://elifesciences.org/articles/62875#fig3video1

stylet bundle is in this lumen (*Figure 3F*). During the feeding process, the stylet bundle protrudes from the apex of the labium (*Figure 4B*, *Figure 3—figure supplement 2A–C*).

The muscles in the labium include three pairs of retractors of the third segment (RT1–RT3), four pairs of lockers of the stylets (LS1–LS4), two pairs of rotators of the second segment (RS1 and RS2), and a pair of rotators of the first segment (RF) (*Figure 3E*, Interactive 3D PDF). RT1 originates from the lateral region of the L2 and inserts into the floor of the labium groove in the L3. RT2 and RT3 both insert into the apodeme of the L3. RT2 originates on the dorsal wall of the L2, while RT3 originates from the lateral wall of the L2. The four pairs of lockers of the stylets have their origins in the dorsal region of the second labium and insert into the floor of the labium groove on the same segment. RS1 and RS2 originate from the dorsal and lateral regions of the L1, respectively, and both insert into the apodeme of L2 (*Figure 3E*, *Figure 3—figure supplement 1C*).

The stylet bundle consists of two mandibular stylets and two maxillary stylets. They emerge from the stylet bases (*Figure 3D*). As the stylets enter the groove, they become adherent to one another forming a compact bundle. The two maxillary stylets are interlocked by longitudinal ridges and grooves (*Figure 3F*). The food canal and the saliva canal are formed where the two maxillary stylets meet. The thicker mandibular stylets lie laterally to the maxillary stylets (*Figure 3F*, *Figure 3—figure supplement 2C,D*). They encircle the maxillary stylets for tight packaging and at the same time allowing themselves to slide along the longitudinal axis.

Every stylet base is enclosed by a cuticular pouch (MdPC and MxPC) that partly wraps the sheath (*Figure 3A,C,D*, *Figure 3—figure supplement 1A*). The mandibular pouch produces two long arms called the mandibular levers (MdL) (*Figure 3A,C,D*). One of the levers proceeds dorsally, and at the end of the lever are the insertion of two retractor muscles. One (RMdA) originates from the dorsal wall of the head capsule, while the other one (RMdB) extends anteriorly. The other lever extends anteriorly to the mandibular plate. This lever supports most of its length a wide and thin epidermal inflection, on which the protractor of the mandibular stylet (PMd) is inserted (*Figure 3A,C*, *Figure 3—figure supplement 1A,B*). The protractor of the mandibular stylet originates in the inner surface of the mandibular plate. The maxillary stylet arises from the maxillary pouch, which is dorsal to its mandibular counterpart. The retractor of the maxillary stylet (RMx) is inserted in the dorsal end of the pouch and has its origin on the dorsal wall of the head capsule, positioned lower than the origin of the RMd (*Figure 3C*). On the lateral side of the pouch, we discerned the insertion point of the protractor of the maxillary stylet (PMx), which originates on the inner surface of the maxillary plate (*Figure 3A*).

## Salivary glands

Sap-sucking requires the function of the salivary glands that are associated with the mouthparts. These paired organ lies in the anterior part of the thorax, lateral to the alimentary tract (*Figure 1A, B*). On each side, a principle gland is connected with an accessary gland by the accessary duct (*Figure 5A*). The efferent salivary ducts arise from the two principle glands and converge anteriorly to form the common salivary duct that transports saliva into the salivary syringe (*Figure 5D–I*).

The principle gland is acinar, with nine kinds of follicles (A–I). It is digitate with the nuclei at the periphery of the cell. Follicles C–I are composed of several compact cells with no clear boundary or interspace between them (*Figure 5—figure supplement 1*). Therefore, cells are difficult to distinguish and count. In follicles A and B, by contrast, the boundaries between the cells are clear. Both include six binucleate cells (A1–A6 and B1–B6) (*Figure 5B*). The follicles are classified according to the density, size, and electron densities of the vesicles in the cells. Among the nine kinds of follicles, follicle D contains the smallest and the highest number of vesicles, while follicle E contains the lowest number of vesicles (*Figure 5—figure supplement 1*). Each one of the accessary glands is composed of two symmetrical follicles, and each follicle contains translucent vesicles (*Figure 5—figure supplement 1J*). Different cell types suggest they may secrete different components of saliva. Ductules from each cell join up into the efferent salivary gland (*Figure 5—figure supplement 1L*). Follicle H is on a separate branch from other follicles. The two branches merge with the accessary duct into the efferent salivary duct, which is a single layer of cells lined with a thin cuticle (*Figure 5—figure supplement 1K*). A nerve bundle protrudes from the ventral pharyngeal sensory center, bypasses the accessary duct, penetrates follicles G and A successively, and then bifurcates to two branches at follicle A, finally they terminates in follicle H (*Figure 5B*).

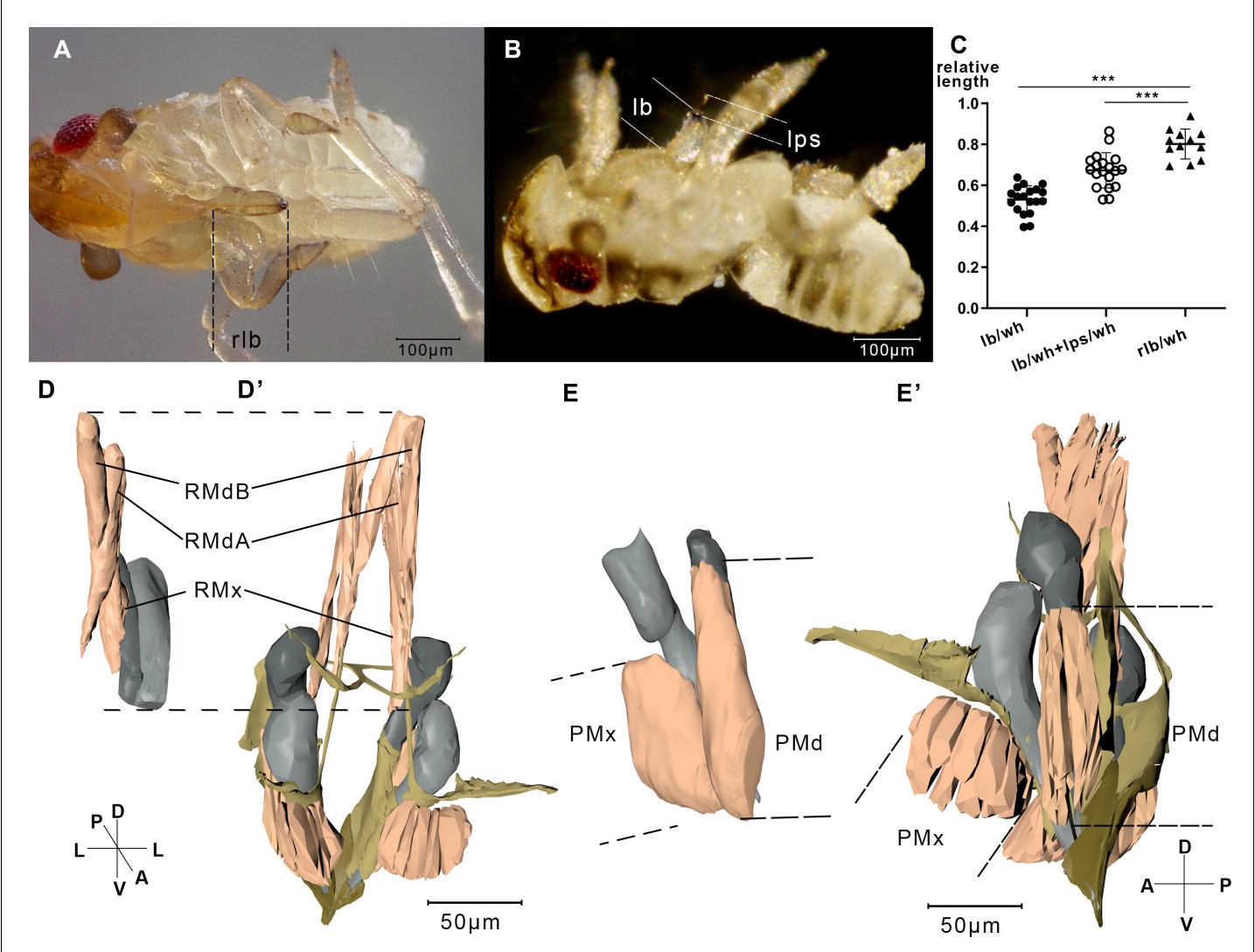

**Figure 4.** Measurement of the labium, stylet, and muscles. (A) A relaxing nymph. Length of the beak (lb) refers to the distance between the tip of the labium and the tip of the labrum. (B) A feeding nymph. Length of the protruding stylet (lps) in feeding nymphs refers to the distance between the tip of the labrum and the tip of the stylet bundle. (C) Relative length of the labium and the stylets in feeding nymphs and relaxing nymphs. lb, lps, and rlb were divided by the width of the head (wh) to eliminate the influence of different body size. rlb, length of the beak in relaxing nymphs. 12 of the relaxing nymphs and 19 of the feeding nymphs were measured. ***, two-tailed t-test, p<0.001. (D) Reconstructed musculature of a feeding nymph (Sample 7). RMdA = 69,720 nm, RMdB = 74,170 nm, RMx = 122,390 nm. See measured lengths and statistic tests in *Figure 4—source data 1*. (D') Reconstructed musculature of a relaxing nymph (sp1). RMdA = 97,550 nm, RMdB = 111,410 nm, RMx = 161,090 nm. (E) Reconstructed musculature of a feeding nymph (sp8). PMd = 65,020 nm, PMx = 85,860 nm. (E') Reconstructed musculature of a relaxing nymph (sp1). PMd = 39,200 nm, PMx = 82,700 nm. All length data of the muscles were obtained using the measurement module in Amira. RMdA and RMdB, retractor of the mandibular stylet A and B; RMx, retractor of the maxillary stylet; PMd, protractor of the mandibular stylet; PMx, protractor of the maxillary stylet. See musculature of feeding nymphs in *Figure 4—videos 1* and *2*. Axis labels are the same as those used in *Figure 1*.

The online version of this article includes the following video and source data for figure 4:

**Source data 1.** Measured lengths of the beaks and stylets.

**Figure 4—video 1.** The volume data of sample 6.

https://elifesciences.org/articles/62875#fig4video1

**Figure 4—video 2.** The volume data of the mouthpart from sample 7.

https://elifesciences.org/articles/62875#fig4video2

The salivary syringe lies at the base of the hypopharynx (*Figure 5A*). Its core component is a pocket-like structure, the salivarium (*Baptist, 1941*), which connects the common salivary duct, the reservoir, and the anterior salivary duct. A pair of dilators of the salivary syringe are attached on the lateral side of the reservoir.

### Cibarial pump

The extreme anterior end of the esophagus, the cibarial pump, is a sclerotized-walled canal with quadrangle cross sections (*Figure 3A*). The cibarial pump lies vertically in the posterior part of the head. Its anterior wall invaginates into the lumen, which renders the lumen of the cibarial pump flexible and easy to expand. The dorsal dilators of the cibarial pump originate from the dorsal wall of the head and insert into the anterior wall of the cibarial pump. The lateral dilators of the cibarial pump insert into the lateral wall of the cibarium pump, with their origins on the anterior arm of the tentorium (*Figure 3A*, *Figure 3—figure supplement 1B*). On the posterior wall of the cibarium pump is the third pair, that is, the posterior dilators of the cibarial pump that have their origins on the posterior arm of the tentorium (*Figure 3B*).

### Contraction in the musculature during feeding process

Only the tip of stylets pierce into the plant tissue during feeding. To figure out how the nymph pulls its stylet out of the labium, we investigated the musculature that might be involved in feeding process on two samples. In feeding modality, unexpectedly, the protractors of the stylets are shortened and retractors are all elongated (*Figure 4D,D', E,E'*).

When the muscles contract to retract the stylet bases, the stylets stretch out of the labium (*Figure 4B*). We observed that adult *N. lugens* lower their heads during the feeding process while the labium tip touches the surface of the plant (*Figure 6—video 1*). We suppose that *N. lugens* might compress its beak to stretch out the stylet. To test this, we measured the beak of several nymphs after freezing them in liquid nitrogen during feeding.

The lengths of the beak (lb) and the protruding stylet (lps) were divided by the width of the head (*wh*) to eliminate a possible influence of different body sizes (*Figure 4A,B*). The lb/wh ratio of the 19 feeding nymphs is $0.530 \pm 0.065$ (mean $\pm$ SEM), significantly smaller than that of the 12 relaxed nymphs with the ratio of $0.801 \pm 0.069$ (mean $\pm$ SEM, *Figure 4C*). The result agrees that the beak is shortened when feeding. The beak is supposed to shorten due to the contraction of the intersegmental muscles in the labium, including the retractors of L1, the rotators of L2, and the rotators of L3. The total length of lb and lps in feeding nymphs divided by wh is $0.673 \pm 0.082$ (mean $\pm$ SEM), significantly smaller than that in relaxing nymphs (*Figure 4C*). Given that in relaxed individuals the stylets and the labium are of the same length, the result suggests that the stylets of feeding nymphs are shorter than those of relaxed nymphs, which agrees with the observation that stylets of feeding nymphs are pulled back by the retractors of the stylets.

## Discussion

### The SBF-SEM technique shows great promise in a broad range

In this study, we applied SBF-SEM to assemble a complete reconstruction of the *N. lugens* nymph with a total length of 600 µm. Compared to micro-CT that was applied to resolve the internal structure of an insect as small as the coffee berry borer (*Hypothenemus hampei*) (*Alba-Alejandre et al., 2019*), SBF-SEM were able to resolve smaller individuals at single cell-scale resolution.

Although the majority of SBF-SEM data are collected in neuroscience studies and the sample preparation methods are optimized for brains and neuronal tissues (*Denk and Horstmann, 2004*; *Tapia et al., 2012*), SBF-SEM is well suitable for entomological studies. SBF-SEM can handle large samples like insects of several hundred microns and small samples like collagen fibrils (*Starborg et al., 2013*). If a higher resolution is desired in a certain sub-volume, SBF-SEM can be combined with some other volume EM imaging technique with higher resolution, such as FIB-SEM and TEMCA (*Guo et al., 2020*; *Xu et al., 2017*; *Zheng et al., 2018*). However, the destructive cutting-and-imaging mode is a drawback when analyzing the sample with various techniques. For example, sensilla on the antenna of an insect can be analyzed by SBF-SEM imaging on the whole antenna, and the following higher resolution imaging on the selected sub-volume can be only conducted on

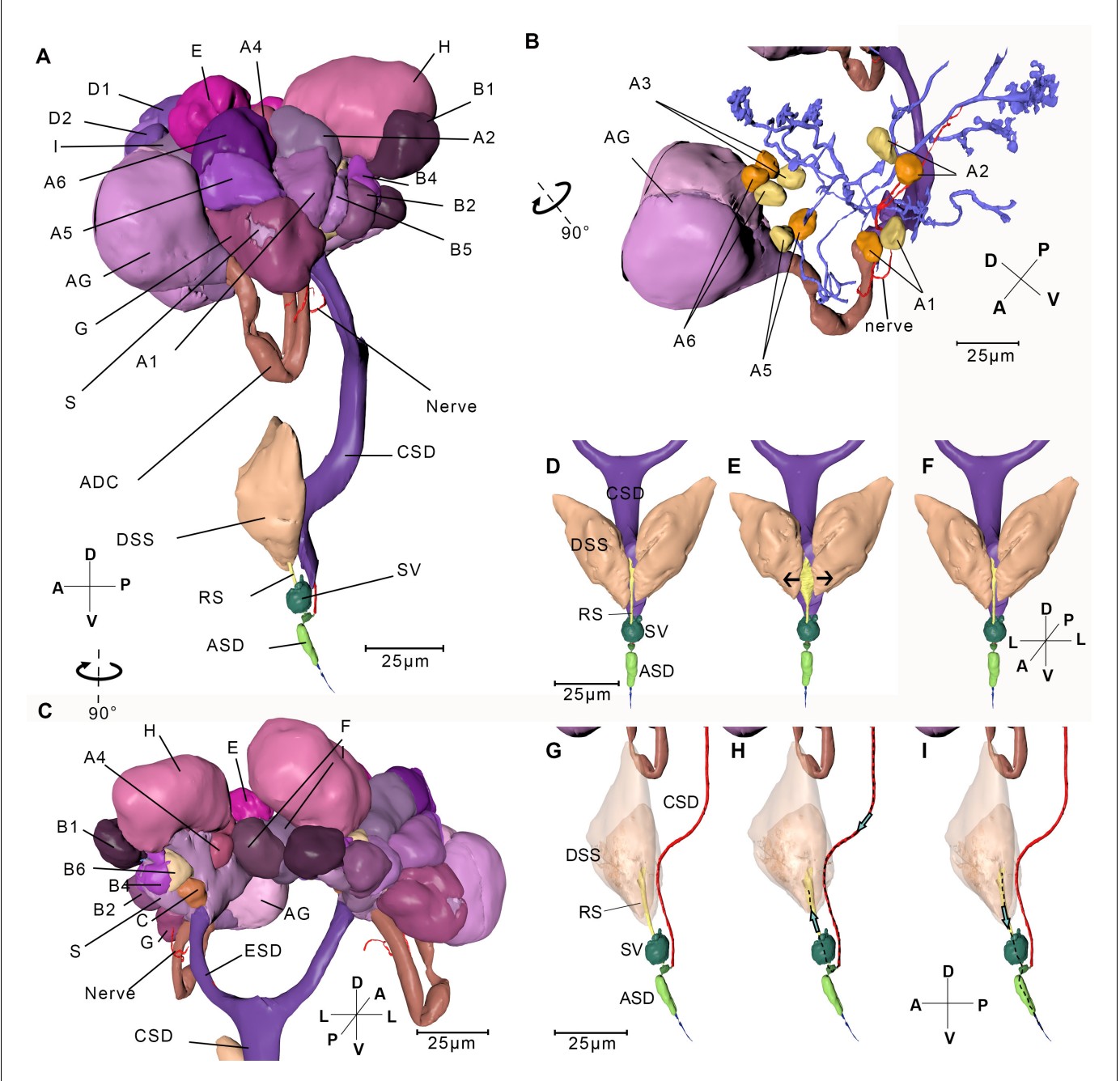

**Figure 5.** Reconstructed salivary gland and the saliva injection process. (**A**) Left-lateral view. A–I, Follicles A–I; ADC, accessary duct; AG, accessory gland; ASD, anterior salivary duct; CSD, common salivary duct; DSS, dilator of the salivary syringe; ESD, efferent salivary syringe; RS, reservoir; S, stroma; SV, salivarium. The serial block-face scanning electron microscopy (SBF-SEM) slices of every follicle are given in *Figure 5—figure supplement 1*. (**B**) Ductules in the left salivary gland. Cells in the principle gland are removed. The two nuclei from each cell in Follicle A are shown in yellow and orange respectively. (**C**) Back view of the salivary gland. (**D–I**) Simulated presentation of the saliva injection process. (**D**) Front view of the salivary syringe. (**E**) A pair of DSS contracts, pulling the wall of the reservoir to the arrowed direction. Consequently, the reservoir expands. (**F**) Dilators of the salivary syringe relax and consequently the reservoir shrinks. (**G**) Lateral view of the salivary syringe. The wall cells of CSD are removed and the lumen is presented. The pair of DSS is rendered transparent. It is in the same state as the specimen of panel (**D**). (**H**) The pair of DSS contract and the reservoir expands. Saliva is pumped into the reservoir. The dotted line indicates the route of the saliva and the arrow indicates the direction of the saliva flux. (**I**) DSS relaxes and the reservoir shrinks, consequently saliva in the reservoir is pumped out. Axis labels are the same as those used in *Figure 1*.
The online version of this article includes the following figure supplement(s) for figure 5:

**Figure supplement 1.** Serial block-face scanning electron microscopy (SBF-SEM) sections showing cells from different follicles in the salivary gland.

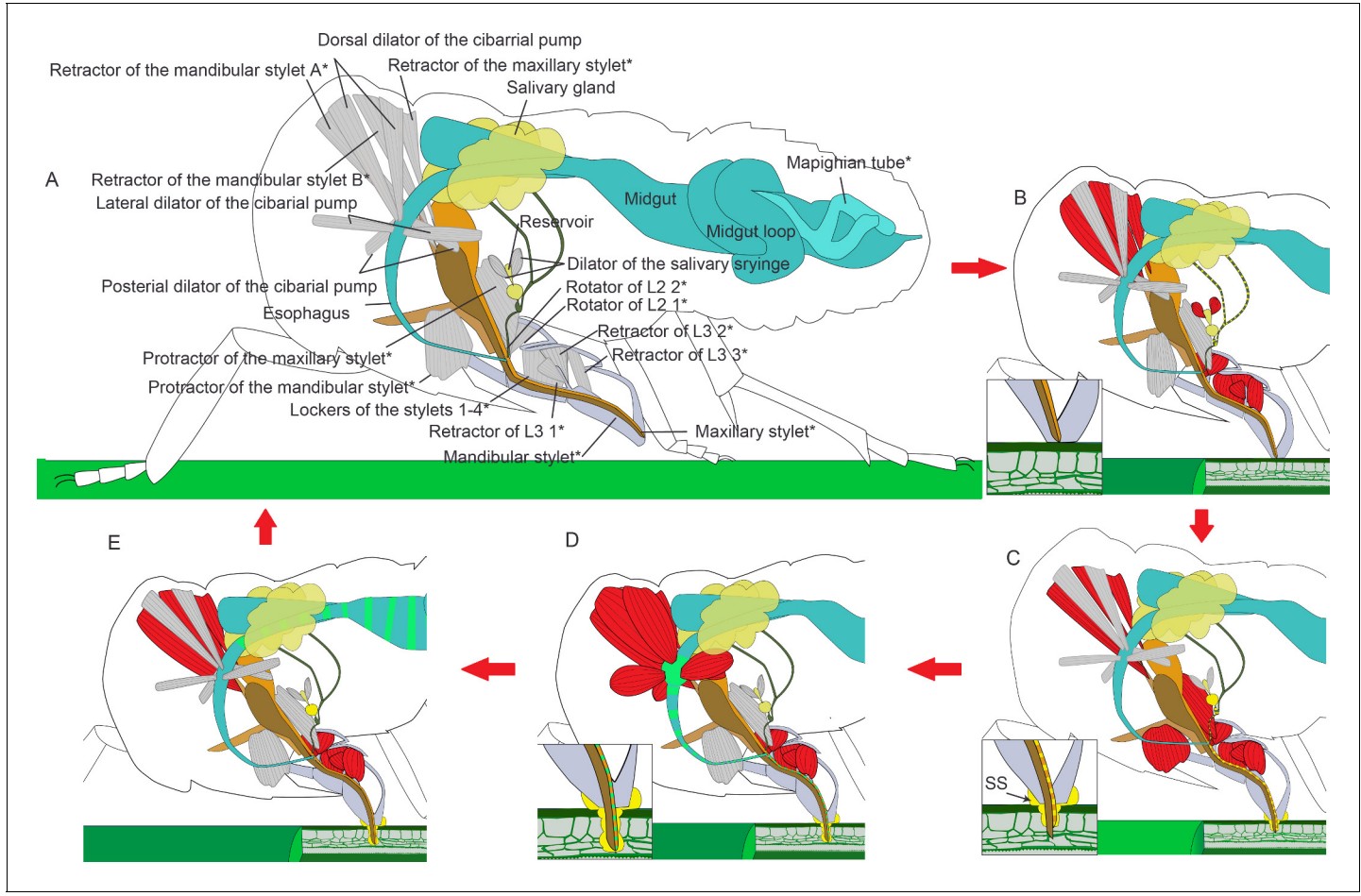

**Figure 6.** Illustrated presentation of a feeding process model. The inserts in panel (**B–D**) show a close view of the stylets tip. Relaxed muscles are in gray and contracting muscles are in red. (**A**) A relaxing insect. The stylet bundle is in the labium. *The structures are paired in the insect, but only one side of the body is shown. (**B**) Rotators of the L2 and retractors of L3 contract, and the labium is bent to the plant. Retractors of the mandibular and maxillary stylet contract, and the stylets are retracted to stay inside of the labium. Dilators of the salivary syringe contract to pump saliva into the reservoir and salivarium. (**C**) Protractor of the mandibular stylet on one side of the body contracts and the corresponding mandibular stylet stick out to dig in the plant. Protractors of the maxillary stylets contract to pull maxillary stylets out. The head lowers down to compress the labium, so the stylets can dig deeper into the plant tissue. Retractors of both the mandibular and the maxillary stylets keep contracting to pull up the mandibular and the maxillary pouch, so that the first segment of the labial can be packed into the head capsule as the head lowers down. Dilators of the salivary syringe relaxes to pump out saliva. Salivary sheath (SS) is formed at the contacting surface of the mouthpart and the plant tissue. (**D**) As the two mandibular stylets penetrate into the plant tissue alternately with the maxillary stylets following, all the four stylets reach the phloem tissue. Dilators of the cibarial pump contract to pump the plant sap into the cibarial pump. (**E**) Dilators of the cibarial pump relax to squeeze the plant sap into the midgut. See feeding process of an adult in *Figure 6—video 1* and an animated feeding process in *Figure 6—video 2*.

The online version of this article includes the following video(s) for figure 6:

**Figure 6—video 1.** The feeding process of an adult *N. lugens*.
https://elifesciences.org/articles/62875#fig6video1

**Figure 6—video 2.** The illustrated animation of the feeding process.
https://elifesciences.org/articles/62875#fig6video2

another sample. The convenient serial sectioning SBF-SEM is also a method of choice for tracking neurons or muscles in studies on 3D structure of connective tissues. Compared with brains and tissue blocks, the waterproof exoskeleton of insects prevents normal fixation and staining. We tried inducing some wounds on the legs to allow rapid fixation and longer staining times to produce better contrasts in images and less damage to the sample.

The full-pipeline of sectioning and imaging can generate large volumes of data automatically, while analyzing the images need tedious manual work. Need of time and experience on analyzing

images can constrain broad usage of SBF-SEM. Further improvements such as deep learning algorithms for image segmentation and better alignment on defective images will reduce manual intervention in the next future.

## Possible feeding process inferred from the musculature clues

We have elucidated the musculature in the labium and the head of *N. lugens* both in relaxing and in feeding modalities. These results helped us to understand muscle movements during the feeding process. Based on reconstruction of the musculature, a novel feeding process model is proposed (*Figure 6*, *Figure 6—video 2*): at the beginning, three pairs of retractors of L3 and two pairs of rotators of L2 contract to bend the labium in a way that the labium tip touches the plant surface vertically. At the same time, it is also shortened by contraction of the intersegment muscles. The retractors of the maxillary stylets and mandibular stylets also contract to withdraw the stylets into the labium and generates space for the first labial segment that is packed into the head capsule (*Figure 6B*). Then the two protractors of the mandibular stylets contract alternately to pull out the stylets to pierce further and deeper into the plant tissue (*Figure 6C*). The labium is compressed both by contracting the intersegment muscles and by packing the first labial segment into the head capsule. At the same time, the insect lowers its head to compress the labium, thereby the stylet bundle sticks further out of the labium. As the two mandibular stylets penetrate into the plant tissue alternately with the maxillary stylets following, all the four stylets reach the phloem tissue (*Figure 6D*). Finally, the head is at a low position, and the first labial segment of the beak is partly retracted into the head capsule. Retractors of the stylets keep contraction to pull up the pouches to generate space for the first labial segment that is packed into the head capsule.

The lockers of the stylets may continue contracting during this process. When they contract, the floor of the labium groove will move dorsally and the lateral side of the labium groove will approach. They are likely to lock the stylet bundle in the groove (*Spangenberg et al., 2013*), so the stylet bundle may be restricted to move along the labium groove.

As the stylets continue penetrating, saliva is secreted into the plant tissue in the following way (*Figures 4D–I* and *6B*). There is no muscle on the salivary duct or the salivary gland, so the salivary syringe is the only possible apparatus for saliva ejection. Its unique structures are modified for actively expelling the saliva. Dilators of the salivary syringe contract to expand the reservoir. Saliva produced in the salivary glands is pumped into the salivarium and the reservoir. When the dilator of the salivary syringe relaxes, the reservoir shrinks and squeezes out the saliva into the anterior salivary duct. The saliva then flows out of the stylets through the saliva canal. As the stylets penetrate into the plant tissue, dilators of the salivary syringe alternately contract and relax to pump out the saliva that includes gel saliva and watery saliva. When the gel saliva is released to the plant surface, it solidifies to generate a saliva interface between the plant surface and the labium tip to immobilize the labium. When it is released into the plant tissue, it diffuses surrounding the stylets and solidifies to a sheath that protects the insect against chemical defenses from the plant (*Huang et al., 2015*; *Sōgawa, 1982*).

The maxillary stylets can dig more into the plant tissue (*Figure 3—figure supplement 2*). There is some evidence that the tip of the maxillary stylet can explore a wide range in the tissue to find the phloem or a blood vessel for the blood-feeding species like Triatominae (Hemiptera: Reduviidae) (*Tull et al., 2020*). Once the maxillary stylets reach the phloem, the sucking process starts. Three pairs of dilator muscles, the dorsal, lateral, and posterior dilator muscles of the cibarial pump, are involved (*Figure 6D,E*). Contraction of muscles lifts the invaginated wall of the cibarial pump, expands the lumen, and consequently creates the upward suction through the food canal. In this manner the liquid food is drawn into the pump lumen l. When the muscles relax, the elastic energy of lifted pump wall is released and it springs back. The lumen is contracted and the liquid food in the pump flows into the esophagus.

The stylets protruding mechanism in hemipteran insects has been discussed in *Homalodisca coagulata* (Membracoidea: Cicadellidae) and diaspidid insects (Coccoidea: Diaspididae) (*Leopold et al., 2003*; *Beardsley and Gonzalez, 1975*). Previously, there is some consensus that the protractor muscles contract to protrude the corresponding stylets (*Leopold et al., 2003*; *Beardsley and Gonzalez, 1975*). In contrast, we found that the protractor muscles did not contract even when the stylet tips were pushed out of the beak, which was beyond previous conception. Protrusion of the stylet resulted mainly by lowering the head and compression of the labium.

**Table 2.** Quantitative analysis of organisms and systems.

| Quantitative analysis | Alimentary canal | | | | | Central nervous system | |
|---|---|---|---|---|---|---|---|
| | Esophagus | Anterior diverticulum | Midgut | Hindgut | Malpighian tubes | Cortex | Neuropils and ganglions |
| Volume (µm$^3$) | 52,832.8 | 205,520.6 | 464,625.2 | 935,172.6 | 72,377.42 | 2,232,803.2 | 1,507,924.3 |
| Volume percentage* | 0.27% | 1.05% | 2.37% | 4.77% | 0.37% | 11.40% | 7.70% |

| Quantitative analysis | Fat body | | Aorta | Trachea | Salivary gland | |
|---|---|---|---|---|---|---|
| | YLS | Thread-like symbiont | | | Accessary gland | Principle gland |
| Volume (µm$^3$) | 824,947.6 | 126,447.7 | 8,192.6 | 23,671.1 | 71,422.9 | 268,774.4 |
| Volume percentage* | 4.21% | 0.65% | 0.04% | 0.12% | 0.36% | 1.37% |

*Volume percentage, the percentage of the organ volume in the body volume. All volumes were calculated using *label analysis* module in Amira.

Although there are few reports on inner musculation, the exterior structures of the mouthparts of the hemipteran species have been studied a lot. The mouthpart structures of the Hemiptera insects are similar, probably due to evolutionary relatedness (e.g. *Dai et al., 2014*; *Mora et al., 2001*). They share similar morphological characteristics, such as a segmented labium that wraps the stylets, interlocking stylets, and two canals in the stylet bundle. Given that function is strictly constrained by structure, we assume that the *N. lugens* feeding mechanism also applies to other hemipteran species.

In some species with bigger body sizes, the labium is bended rather than compressed. The seed bug *Pyrrhocoris sibiricus* (Pyrrhocoroidea: Pyrrhocoridae) and the stink bug *Erthesina fullo* (Pentatomoidea: Pentatomidae) are reported to bend their labium between the first segment and the second segment to protrude the stylet (*Wang and Dai, 2017*; *Wang and Dai, 2020*). These works did not mention if the head was lowered down. We reckon that the head should be lowered in this situation. The bent labium shortens the straight-line distance between the base and the tip of the labium. Theoretically, when the labium bends, the tip of the labium does not reach the surface of the plant unless the head lowers down. It may be common in this family that the head-lowering behavior and bending of the labium are combined when protruding the stylets. The head-lowering behavior in aphids is combined with lifting of the abdomen. They even position their body nearly vertical to the plant surface when sucking in sap (*Guerrieri and Digilio, 2008*). Aphids do not bend their labium, so they probably compress the labium as in the case of *N. lugens*. Scale insects, like *Paraputo guatemalensis* (Coccoidea: Pseudococcidae) and *P. odontomachi*, have stylet bundles that are much longer than the labium. They usually coil the stylet bundle and hide it in the head capsule (*Beardsley and Gonzalez, 1975*), so they are unable to protrude the stylet bundle by lowering the head. An aphid species also have extremely long mouthparts, and their labium can dig into the plant tissue (*Brożek et al., 2015*). Therefore, they probably adopt a different way of piercing their mouthparts into the plant. Overall, the planthopper stylet penetration mechanism might apply to most sap-sucking hemipterans with short stylets.

## The alimentary canal shows water-disposal mechanism

The filter chamber locates in the alimentary canal of most hemipteran insects that feed on fluid food. It connects the anterior and the posterior parts of the midgut and transfers extra water directly from the anterior to the posterior end of the midgut. Without excess water, the sap in the midgut is concentrated 10-fold (*Terra and Ferreira, 2012*). This water-disposal mechanism may quickly expel water, ions, and soluble sugars to the hindgut, while amino acids, proteins, and lipids will be retained and digested in the midgut (*Marshall and Cheung, 1974*; *Salvucci et al., 1998*). However, a filter chamber is not present in *N. lugens*, and a different water-disposal mechanism is proposed.

There are some aphid families such as Aphididae, Adelgidae, Mindaridae, and Chaitophoridae without a filter chamber (*Douglas, 2003*). They develop coiled segments of the hindgut adhering to the stomach. *Shakesby et al., 2009* revealed that *in the pea aphid Acyrthosiphon pisum (Aphidoidea: Aphididae) the stomach and distal intestine are rich in water-transporters called aquaporins. The hypothesis is that water is transferred from the stomach to the distal intestine through the*

**Table 3.** Nomenclature system and abbreviations of the structures.

| Central nervous system | | Alimentary canal | | Mouthpart | |
|---|---|---|---|---|---|
| Abdominal ganglion | AG | Anterior diverticulum | AD | Base of the mandibular stylet | BMd |
| Abdominal nerve | AbN | Dorsal dilator of the cibarial pump | DDCP | Base of the maxillary stylet | BMx |
| Antennal lobe | AL | Esophagus | ESO | Food canal | FC |
| Antennal mechanosensory and motor center | AMMC | Hindgut | HG | Length of the beak | lb |
| Antennal nerve | AN | Lateral dilator of the cibarial pump | LDCP | Length of the protruding stylet | lps |
| Anterior optic tubercle | AOTU | Malpighian tubule | MT | Lockers of the stylets 1–4 | LS1-4 |
| Calyx | CA | Midgut | MG | Protractor of the mandibular stylet | PMd |
| Cornea | CN | Midgut loop | ML | Protractor of the maxillary stylet | PMx |
| Cortex | COR | Posterior dilator of the cibarial pump | PDCP | Retractor of the mandibular stylet A | RMdA |
| Central complex | CX | | | Retractor of the mandibular stylet B | RMdB |
| Flange | FLA | **Cephalic endoskeleton** | | Retractor of the maxillary stylet | RMx |
| Inferior neuropil | INP | Depressor of the trochanter | DT | Retractors of L3 1–3 | RT1-3 |
| Lamina | LA | Corpotentorium | CPT | Rotator of L2 1–2 | RS1-2 |
| Labial nerve | LBN | Mandibular levers | MdL | Rotators of L1 | RF |
| Labial sensory center | LBS | Mandibular pouch | MdPC | Saliva canal | SC |
| Proleg nerve | LN1 | Maxillary pouch | MxPC | Stylet bundle | SB |
| Mesoleg nerve | LN2 | Maxillary plate | MxP | The first labial segment | L1 |
| Metaleg nerve | LN3 | Mandibular plate | MdP | The second labial segment | L2 |
| Lobula | LO | | | The third labial segment | L3 |
| Lobula plate | LP | **Salivary gland** | **SG** | Width of the head | wh |
| Mushroom body | MB | Accessory duct | ADC | | |
| Medulla | MED | Accessory gland | AG | **Treacheal System** | |
| Medial lobe | ML | Anterior salivary duct | ASD | Atrium | *str* |
| Neuropil | NP | Common salivary gland | CSD | Dorsal longitudinal trunk | *dlt* |
| Optic lobe | OL | Dilator of the salivary syringe | DSS | Dorsal longitudinal trunk from ts1 to *ts2* | *dlt1* |
| Olfactory sensory neuron | OSN | Efferent salivary gland | ESD | Dorsal longitudinal trunk from ts2 to *as1* | *dlt2* |
| Protocerebral bridge | PB | Reservoir | RS | Mesoleg trunk | *mlt* |
| Pedunculus | PED | Salivarium | RV | Spiracle on the 1st-8th abdominal segments | *as1-8* |
| Prow | PRW | Stroma | S | Spiracle on the mesothoracic segment | *ts1* |
| Saddle | SAD | | | Spiracle on the metathoracic segment | *ts2* |
| Superior lateral neuropil | SLNP | **Symbionts** | | Spiracular valve | *spv* |
| Superior neuropils | SNP | Fat body | FB | The second trachea originating from *ts1* extending posteriorly | *pt2* |
| Superior posterior slope | SPS | Yeast-like symbionts | YLS | The valve separating the atrium | *v2* |
| Prothoracic ganglion | TG1 | Thread-like symbionts | TLS | The trachea extending anteriorly from *ts1* | *at* |
| Mesothoracic ganglion | TG2 | | | The trachea extending posteriorly from *ts1* | *pt1* |
| Metathoracic ganglion | TG3 | **Compound eye** | **CE** | Thorax lateral trunk | *tlt* |
| Ventral nerve cord | VNC | Crystalline cone | CC | | |
| Ventral pharyngeal sensory center | VPS | Rhabdom | Rh | | |
| | | Semper cell | SC | | |

*activity of aquaporins at the point where the hindgut contacts the stomach. In N. lugens*, we also found a similar structure. The distal end of the anterior midgut contacts the anterior midgut (**Figure 2—figure supplement 1E**). The epithelia of the anterior midgut become thin at this point, which may facilitate water in the midgut to move to the end of the midgut loop through the epithelium.

In the anterior part of the midgut of *N. lugens*, protuberance on the inner surface may enable the gut lumen to increase by expanding the folds. In contrast, the loop region narrows and develops dense microvilli to increase the absorption surface and the time that food travels through the midgut. When phloem sap is ingested, it moves quickly into the anterior part but slowly into the loop region. As a result, most of the sap may stop in the anterior part, and the gut lumen expands to temporarily store food. At the distal end of the dilated anterior part, the epithelium becomes thinner than at neighboring positions at a point where the end of the midgut loop contacts it. At this place, water in the sap may be transported to the end of the loop region through the epithelia, thereby concentrating proteins and lipid that may then move into the anterior part of the midgut loop for further digestion and absorption. This may be a *N. lugens* way for concentrating nutriments in the food.

The order Hemiptera consists of four higher taxa as Heteroptera (red bugs, water bugs, seed bugs, lace bugs, bedbugs, and stinkbugs), Sternorrhyncha (aphids, scale insects, whiteflies, and psyllids), Auchenorrhyncha (cicadas, froghoppers, leafhoppers, and planthoppers), and Coleorrhyncha (moss bugs) (**Johnson et al., 2018**; **Figure 2—figure supplement 5**). Almost all members of this order rely on a fluid diet, but the morphology of the alimentary canals varies a lot. Their diverse water-disposal mechanisms give some clues in alimentary canal evolution. Species of Pentatomomorpha in the suborder Heteroptera have a discontinuous alimentary canal so that the fluid food does not pass through the midgut. In these species, the intestinal symbiotic organ blocks food flow from the anterior part of the midgut by an extreme narrow region. Food fluids in the anterior regions of the midgut are completely absorbed and transferred into the hemolymph and excreted through the Malpighian tubules back into the hindgut (**Ohbayashi et al., 2015**). Other families of Heteroptera including herbivorous Cimicomoepha and Leptopodmorpha, predatory Nepomorpha and Gerrmorpha have continuous intestines showing no unique water-disposal mechanism (**Habibi et al., 2008**; **Nardi et al., 2019**; **Rost-Roszkowska et al., 2017**). Their convoluted alimentary canals indicate osmotic water exchanges by juxtapositioning an anterior region of the midgut to the most posterior region as in *N. lugens*. Members of the suborder Auchenorrhyncha have filter chambers with exception of those species belonging to Fulgoroidea (**Hickernell, 1923**; **Kershaw, 1914**; **Zhong et al., 2015**). This indicates a simplification of the alimentary canal in an ancestor of Fulgoroidea. Members of the suborder Sternorrhyncha have a filter chamber, except for most families of Aphidoidea that develop coiled segments of the hindgut adhering to the stomach, as mentioned above (**Kruse et al., 2017**; **Mathew et al., 2011**; **Peeters et al., 2017**). Members of *Coloradoa* and *Hyalopterus* in Aphidoidea still have filter chambers, so loss of the filter chamber occurred only in some families (**Ponsen, 1991**). The ancestors of Fulgoroidea and Aphidoidea experienced similar morphological transformations. This may reflect the impact of similar diet and life style on evolution.

## Yeast-like symbionts and other endosymbionts

The association between insects and endosymbionts is widely found in nature and in most cases the endosymbionts are bacteria (**Wernegreen, 2002**). The phenomena that symbionts synthesize and supply amino acid for their hosts are well known in hemipteran insects. The aphid bacterial symbiont (*Buchnera aphidicola*) synthesizes essential amino acids (**Shigenobu et al., 2000**). The leafhopper (*Dalbulus maidis*) symbiont *Nasuia* has the smallest bacterial genome and provides two essential amino acids to the host (**Bennett and Moran, 2013**). *Burkholderia* symbionts found in the species of the superfamilies Lygaeoidea and Coreoidea have been shown to enhance host innate immunity, but do not have any specific nutritional function (**Kikuchi et al., 2011**; **Kim et al., 2015**).

*N. lugens* is an exception insofar as a fungus is the dominant endosymbiont. The yeast-like fungus is an obligate endosymbiont in the planthopper. They cannot survive in vitro and usually reside in the fat body mycetocytes, though in this case, we found some of them in legs and heads. *Xue et al., 2014* reported that *N. lugens* lacks the ability to synthesize essential amino acids that are missing in its diet, the rice phloem sap (**Hayashi and Chino, 1990**). *Fan et al., 2015* revealed that genes involved in the synthesis of essential amino acids and cholesterol are found in the genome of the yeast-like symbionts but not in the BPH genome, indicating that the yeast-like symbionts provide

essential amino acids and cholesterol to their host. Yeast-like symbionts are maternally transmitted to the eggs and are present in all developmental stages of the host insect (*Cheng and Hou, 2001*). We found yeast-like symbionts in a single huge mycetocyte that occupied 22% of the abdominal volume. The mycetocyte is likely to grow bigger as the symbionts thrive, but it does not undergo a cell division during the embryonic and nymphal stages.

The anterior diverticulum in hemipterans is often filled with air bubbles; however, its function remains controversial. The Cercopid *Tomaspis saccharina* (Cercopoidea: Cercopoidea) takes in air with the fluid food and excretes air bubbles coated with raucinoid through the anus to produce froths, so AD is likely to separate the air from the food when the fluid passes down into the midgut (*Kershaw, 1914*). Kershaw also proposed that the AD worked as a reservoir to store innutritious fluid and wax materials separated from the food. On the other hand, Goodchild found that the AD contains air at the time of moulting and suggested that the AD inflates the thorax during molting (*Goodchild, 1966*).

Despite these speculations, the AD might serve as a specialized organ to host symbionts in *N. lugens*. Numerous bacterial symbionts that are much smaller than the yeast-like symbionts are in the wall of the AD. They reside in the hollow AD cells. Our 16 s sequencing results showed that the dominant microbe in the AD is Arsenophonus species, which is reported to kill the male eggs in the wasp, *Nasonia vitripennis* but in *N. lugens* shows effect on host insecticide resistance (*Pang et al., 2018*). In previous studies, Arsenophonus was identified as the second most abundant symbiont after the yeast-like symbionts in *N. lugens.* It mainly resides in the fat body (*Fan et al., 2016*; *Ly et al., 2013*). In this case, we believe that Arsenophonus species reside also in the anterior diverticulum cells.

The thread-like symbionts in fat body mycetocytes and the symbionts in the AD wall have not been reported in precedent publications. They reside in certain cells in the body instead of being uniformly distributed in the abdomen like the yeast-like symbionts. Therefore, it is difficult to find them in sections in conventional TEM-based studies. We were able to find them probably because we collected serial sections of the entire nymph. The mycetocytes that host the thread-like symbionts lie very close to the midgut. They may be involved in nutrient absorption. Further studies are expected to unravel the relationship between the insect and these intracellular microorganisms.

## The four-way tracheal ring facilitates efficient gas change

Early studies on the anatomy of the *Drosophila* and Coleopteran tracheal systems are based on dissections and hand-made illustrations (*Whitten, 1957*; *Tonapi, 1978*). Given that thin tracheae are easily neglected and hand drawing are error-prone, the results are not reliable enough, and quantitative descriptions are difficult. The µCT system provides more reliable and complete results allowing elaborate comparisons (*Iwan et al., 2015*; *Socha et al., 2010*). The volume of total tracheae in *N. lugens* accounted for about 0.12% of the body volume. Compared to the tracheal density of 0.5% in *Tribolium castaneum* (Coleoptera: Tenebrionidea), 4.8% in *Eleodes obscura* (Coleoptera: Tenebrionidea) calculated using the stereological point count method (*Kaiser et al., 2007*), and 0.6% in *Tenebrio molitor* (Coleoptera: Tenebrionidea) larva (*Raś et al., 2018*), the tracheal density in *N. lugens* is relatively low. This is in agreement with the hypothesis that larger insects invest more on the tracheal system (*Callier and Nijhout, 2011*; *Kaiser et al., 2007*).

The planthopper has a simplified tracheal system without ventral longitudinal trunks and air sacs that are often found in insects with well-developed flight ability (*Gunn, 1931*). There is no study on tracheal systems in small insects, so it is unclear whether we can generalize our simplification. *N. lugens* excels at long-distance movement. Every spring the migration starts from the Indochina peninsula to eastern China, Japan, and Korea with southwesterly monsoons (*Bao et al., 2000*; *Sogawa and Cheng CH, 1979*). Obviously, a simple tracheal system is able to satisfy the need of oxygen during flight. This may be partly due to the small body size of *N. lugens* that allows efficient oxygen diffusion. In addition, the novel structure described here may also facilitate gas exchange.

The four-way tracheal ring that connects nearby four spiracles is a structure found in insects for the first time. Airflow in the ring is possible both longitudinally and transversely. It is likely to compensate for the lack of ventral longitudinal trunks. The dorsal and ventral longitudinal trunks are present in many other insects. Generally, the airflow in the insect body is longitudinal, which means that the air is taken in through the anterior spiracles and expelled though the posterior spiracles as in the cockroaches (*Heinrich et al., 2013*) and the locust (*Miller, 1960*), or in the reverse direction as

in the wingless dung beetles (*Duncan and Byrne, 2002*). If air flows in and out though the same spiracle, the incoming air will mix with the used air and the gas exchange efficiency will decrease. The longitudinal and transverse trunks that connect different spiracles allow airflow between spiracles, resulting in more efficient gas exchange. In *N. lugens*, oxygen diffuses longitudinally at the dorsal side through the dorsal longitudinal trunks, while at the ventral side, there are no longitudinal trunks. Instead, oxygen diffuses longitudinally through the four-way tracheal rings to efficiently supply oxygen to many tissues located in the thorax and abdomen.

## Conclusion

The BPH, a member of the superfamily Fulgoroidea, belongs to the dominant group of phytophagous Hemipterans that exceeds 12,500 described species. Many species in Fulgoroidea are economically significant pests of major agricultural crops for their direct wounding of plants, high reproductive ability, long-distance migration, and status in virus transmitting. Therefore, detailed studies of the internal structure and feeding mechanism are essential to support research on pest control.

In this study, we report on a 3D reconstruction of a whole *N. lugens* nymph and its sap-sucking mechanism for the first time using SBF-SEM. The new technology is well suitable for a tiny insect and yielded many details with high resolution. We present a set of new findings on the internal structures including 24 neuropils in the central nerve system, nine kinds of follicles in the principle salivary gland with different cell numbers and vesicles, an admirable structure of four-way tracheal rings connecting the spiracles in adjacent segments, fungal endosymbionts in a single huge mycetocyte occupying 22% of the abdominal volume, and symbionts in the crypts of the anterior diverticulum. We also provide videos and interactive three-dimensional PDF versions of the reconstructed structures, which may serve as comprehensible tools for scientists to explore the internal structures of an insect or for teaching.

Based on the 3D reconstruction of the cephalic musculature and movement of all feeding apparatuses, we propose a novel feeding model for the phloem sap-sucking mechanism that may also be applied for many other species with similar mouthparts. These results allow an integral understanding of the actual position and relationships of the structures and organ systems in a tiny insect, especially during the food ingestion and digestion, endosymbiosis, and respiration. The information will serve to study how a tiny pest injures its plant host, and the structures as adaptations shaped in the long history of natural selection. The piercing mechanism may even inspire us to new ideas of tunnel excavation in bionics. Further studies on the indispensable relationship of different symbionts with the host may be valuable in developing microbial pesticides.

# Materials and methods

**Key resources table**

| Reagent type (species) or resource | Designation | Source or reference | Identifiers | Additional information |
|---|---|---|---|---|
| Software, algorithm | GraphPad Prism 8 | GraphPad Prism | RRID:SCR_002798 | |
| Software, algorithm | Adobe Illustrator | Adobe Inc | RRID:SCR_014198 | |
| Software, algorithm | Amira 6.8 | Thermo Fisher Scientific | RRID:SCR_007353 | |

## Insects

The BPHs used in this study were originally collected in Hangzhou (30°16′N, 12°11′E), China. They were reared at 26 ± 0.5°C on rice seedlings under a 16:8 hr (light: dark) photoperiod.

## Samples and preparation

We prepared a first instar nymph (Sample 1) and collected 12,000 digital images of the whole body (*Figure 1—video 1*) at a resolution high enough to reconstruct the tracheae and muscles. A head with the prothorax (Sample 2, *Figure 1—video 2*), a head with intact mouthpart (Sample 3), an abdominal part with several segments (Sample 4), and a few segments on the posterior end (Sample 5) of other first instar nymphs were prepared to collect digital images at similar resolution as Sample

1. Images from the whole body were used for reconstruction and the other images provided additional information for locating structural features.

To study muscle contraction during the feeding process, the first instar nymphs were immobilized with liquid nitrogen while they were sucking on rice plants. They were frozen fast ensuring that the musculature remained in the feeding state. The insects were then removed from the plant and fixed in 2.5% glutaraldehyde (TED PELLA, Lot No.: 2171002) at room temperature immediately (Sample six and Sample 7, *Figure 4—video 1*, *Figure 4—video 2*) Detailed samples description and image resolution are listed in *Table 1*.

The samples were fixed quickly for 24 hr in 2.5% glutaraldehyde (TED PELLA, Lot No.: 2171002), and 0.003% CaCl$_2$ (Sigma, C-2661–500G) in sodium cacodylate (Sigma, CAS: 6131-99-3) buffer (0.1 M). The tissue blocks were then washed in sodium cacodylate buffer (0.1 M) and treated with a solution containing equal volumes of 2% OsO$_4$ (TED PELLA, Lot No: 4008–160501) and 3% potassium ferrocyanide (Sigma, CAS: 14459-95-1) in 0.3 M sodium cacodylate with 4 mM CaCl$_2$ for 1 hr on ice. After rinsing with double-distilled water (ddH$_2$O), the samples were incubated in 1% thiocarbohydrazide (Sigma, CAS: 2231-57-4) (in water) for 20 min at room temperature. Then, the samples were rinsed with ddH$_2$O and treated with 2% aqueous OsO$_4$ for 30 min at room temperature. The tissue blocks were washed in ddH$_2$O and immersed overnight at 4°C in 1% aqueous uranyl acetate. After washing in ddH$_2$O, the samples were incubated in 0.66% lead nitrate (Sigma, CAS: 10099-74-8) diluted in 0.03 M L-aspartic acid (Sigma, CAS: 56-84-8) (pH 5.5) for 30 min at 60°C, then dehydrated in an ascending ethanol series and flat-embedded in EPON 812 resin (EMS, cat. no: 14900) for 48 hr at 60°C.

## Data collection

Resin blocks were carefully trimmed using a Leica EM trimmer, until the surface of the black tissue in the block could be observed. Next, the resin blocks were glued on a stub with electrically conductive colloidal silver (TED PELLA, Lot NO: 169773610), followed by coating with about 30 nm platinum using a sputter coater (Leica, ACE200). 3D data was obtained by a scanning electron microscope (Thermo Fisher, Teneo VS) with one ultramicrotome in the specimen chamber, which allowed synchronous sectioning of resin blocks and imaging of the sample surface. Section thickness and pixel size for each sample are listed in *Table 1*. Each serial face was imaged with 2.5-kV acceleration voltage and 0.2-nA current in backscatter mode with a VS DBS detector. The image store resolution was set to 6144 × 6144 pixels with a dwell time of 2 μs per pixel.

## Image processing and segmentation

The images were aligned, filtered, manually segmented, and used to generate surfaces in Amira 6.8 (Thermo Fisher Scientific).

## 3D reconstruction and presentation

The surfaces generated in Amira were reduced to 10% and then exported as Wavefront .obj files. Maxon Cinema 4D R19 (Maxon Computer GmbH) was used to reassemble the structures, and the segmentation artifacts were carefully removed using the sculpting tool. The resulting C4D projects were used in the video (*Figure 1—video 3*) and static images were exported. For the interactive 3D model, the polygon counts were further reduced to 10% or 1% according to the details of the structures. The simplified C4D project was exported as .3ds file, colored, renamed, saved as .u3d file in Deep Exploration 6 (Right Hemisphere), and embedded into the interactive PDF file using Acrobat Pro DC (Adobe) (Supplementary file). Volumes of all reconstructed structures were calculated using the *label analysis* tool in Amira (*Table 2*). The length of muscles was measured using the measurement tool in Amira.

## Measurement and statistical analysis

To measure the length of the beak, the first instar nymphs were fast frozen by liquid nitrogen when they were sucking in phloem sap on a rice plant. The nymphs were warmed up to room temperature and carefully removed from the plant. A stereoscope was used to measure the distance between two eyes as the width of the head (wh), the length of the beak, and the length of the protruding stylets. The length of the beak (lb) refers to the distance between the tip of the labrum and the tip

of the labium, and the length of the protruding stylets (lps) refers to the distance between the tip of the stylet and the tip of the labium. The measurements are presented as mean ± SEM. Statistical analyses were performed using a two-tailed Student's t-test. p-values <0.001 were considered as statistically significant.

### Conventional SEM and cryo-SEM

For conventional SEM (CSEM), nymphs feeding on rice plants were fixed in liquid nitrogen and post fixed in glutaraldehyde (2.5% with 0.1 M phosphate buffer, at pH 7.4, overnight at 4°C). Then the samples were washed in the phosphate buffer (pH 7.4) and dehydrated in an ascending ethanol series (30%, 50%, 70%, and 90%, followed by 3*100%). After critical drying with an automated critical point dryer (Leica, CPD300), the samples were mounted on double-sided carbon tape on stubs. They were then plasma coated with 10 nm gold using a sputter coater (Leica, ACE200) and viewed under a field emission scanning electron microscope (Thermo Fisher, Helios UC G3). The image was used in *Figure 3—figure supplement 2C*.

For cryo-SEM, nymphs feeding on rice plants were fixed in liquid nitrogen and transferred to a Deben Cryo-SEM system which preserved samples at −20°C. Samples were examined under a field emission scanning electron microscope (ZEISS, EVO15) at low vacuum mode (70 Pa). The images were used in *Figure 3—figure supplement 2A,B*.

### Terminology

For structures whose homologous counterparts were reported in other insects, the authors would prefer using the existed terminological systems instead of starting de novo. Nomenclature of neuropils followed *Insect Brain Name Working Group et al., 2014*, and the external sclerites followed that of *Wipfler et al., 2016*. Muscles in the labium are continuously numbered in order of appearance and follow the terminology established in *Spangenberg et al., 2013*. Abbreviations of the structures are presented in *Table 3*.

## Acknowledgements

This work was supported by the National Natural Science Foundation of China (31630057 and 31871954) and the Natural Science Foundation of Zhejiang Province (LQ20C040003). The EM data were collected at the Center of Cryo-Electron Microscopy, Zhejiang University.

## Additional information

### Competing interests

Yang Yu: Yang Yu is affiliated with Carl Zeiss (Shanghai) Co., Ltd. The author has no financial interests to declare. The other authors declare that no competing interests exist.

### Funding

| Funder | Grant reference number | Author |
|---|---|---|
| National Natural Science Foundation of China | 31630057 | Chuan Xi Zhang |
| National Natural Science Foundation of China | 31871954 | Chuan Xi Zhang |
| Natural Science Foundation of Zhejiang Province | LQ20C040003 | Jian-sheng Guo |

The funders had no role in study design, data collection and interpretation, or the decision to submit the work for publication.

### Author contributions

Xin-Qiu Wang, Formal analysis, Investigation, Writing - original draft; Jian-sheng Guo, Conceptualization, Investigation, Methodology, Writing - review and editing; Dan-Ting Li, Investigation; Yang

Yu, Methodology; Jaco Hagoort, Software, Methodology; Bernard Moussian, Writing - review and editing; Chuan-Xi Zhang, Conceptualization, Supervision, Funding acquisition, Writing - review and editing

### Author ORCIDs
Xin-Qiu Wang (iD) https://orcid.org/0000-0003-2961-1211
Bernard Moussian (iD) http://orcid.org/0000-0002-2854-9500
Chuan-Xi Zhang (iD) https://orcid.org/0000-0002-7784-1188

### Decision letter and Author response
Decision letter https://doi.org/10.7554/eLife.62875.sa1
Author response https://doi.org/10.7554/eLife.62875.sa2

## Additional files
### Supplementary files
- Supplementary file 1. Interactive 3D PDF for the planthopper model.
- Transparent reporting form

### Data availability
All data generated or analysed during this study are included in the manuscript and supporting files. Source data files have been provided for Figures 1–5.

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
