## [Decision Letter]

**Acceptance summary:**

Hemipteran (true bugs) insects including the Chagas disease transmitting kissing bug and plant-pathogenic aphids have specialized mouthparts to suck in liquid food. In this paper, the authors use electron microscopy to reconstruct in detail the structure of these minuscule mouthparts and examine how the morphology matches function. It's an elegant study that should be an intriguing read for anyone interested in animal morphology and evolution.

**Decision letter after peer review:**

Thank you for submitting your article "Three-dimensional reconstruction of a whole insect reveals its phloem sap-sucking mechanism at nano-resolution" for consideration by *eLife*. Your article has been reviewed by three peer reviewers, and the evaluation has been overseen by Michael Eisen as the Senior and Reviewing Editor. The following individual involved in review of your submission has agreed to reveal their identity: Bruno Humbel (Reviewer #3).

The reviewers have discussed the reviews with one another and the Reviewing Editor has drafted this decision to help you prepare a revised submission.

Summary:

In this paper, the authors provide a nanoscale reconstruction of the internal structures of an insect for the first time. They used the brown planthopper as a model and also identified the morphological and mechanical basis of feeding mechanism (that they gained by freezing in liquid nitrogen feeding nymphs). Several novel structures or details of previously known structures are identified. I agree that this could be seen as quite useful for textbooks, for the field of entomology in general, and could help push a renaissance in the kind of basic biology that is so rare to see at the organismal-level these days.

Essential revisions:

1) The greatest challenge for this manuscript is making it relevant to the wider audience of *eLife*, and it is incumbent on the authors to highlight the reasons why it should appeal to a broad audience and not just entomologists. The results are fascinating, but it needs to be laid out why they matter more generally. The importance should be made more explicit – the authors touch upon it briefly in the Introduction (e.g. true bugs can injure crops, pest plant). However, the Discussion should elaborate upon how the results could inform these aspects, such as crop protection, or even physiology or evolution, and should really emphasize the novelty of the technique and what it could mean going forward. All this to say, the broader impacts should be better emphasized. The Discussion, at present, isn't so much an expansion upon the findings as a summary of the results.

2) In the Abstract, Results and Conclusion, the authors mentioned that they found many novel and fascinating internal structures. But in the Discussion, no points were discussed about the new structures. Adding the differences of the internal structures between *N. lugens* and other hemipteran insects, discussing the possible biological function of these novel structures, establishing the relationship between the structure and the life habits of *N. lugens* will be helpful to understand the novel structures.

---

## [Author Response]

Essential revisions:1) The greatest challenge for this manuscript is making it relevant to the wider audience of eLife, and it is incumbent on the authors to highlight the reasons why it should appeal to a broad audience and not just entomologists. The results are fascinating, but it needs to be laid out why they matter more generally. The importance should be made more explicit – the authors touch upon it briefly in the Introduction (e.g. true bugs can injure crops, pest plant). However, the Discussion should elaborate upon how the results could inform these aspects, such as crop protection, or even physiology or evolution, and should really emphasize the novelty of the technique and what it could mean going forward. All this to say, the broader impacts should be better emphasized. The Discussion, at present, isn't so much an expansion upon the findings as a summary of the results.

We thank editors and the reviewers for the suggestions. Throughout the revised manuscript, we have added additional discussion on the importance of our work including the novelty and future expectations of the technique, possible functions of the newly found structures and how they may serve for adaptation of the brown planthopper to various natural environments in the long history of evolution.

2) In the Abstract, Results and Conclusion, the authors mentioned that they found many novel and fascinating internal structures. But in the Discussion, no points were discussed about the new structures. Adding the differences of the internal structures between N. lugens and other hemipteran insects, discussing the possible biological function of these novel structures, establishing the relationship between the structure and the life habits of N. lugens will be helpful to understand the novel structures.

We expanded the Discussion in the following two aspects: (1) possible biological functions of novel structures including the four-way tracheal ring, the anterior diverticulum, the water-excretion mechanism and how they may allow *N. lugens* adapting to natural habitats; (2) comparison of the alimentary canal systems and stylet protruding mechanisms between *N. lugens* and other Hemipteran.